# Effects of Bionic Curves on Penetration Force under Difference Soils

**Yunhai Ma** [1,2,3]**, Huixin Wang** [1,2,3]**, Jian Zhuang** [1,2,3,*]**, Hongyan Qi** [1,2,3] **and Jiangtao Yu** [1,2,3]

1   The College of Biological and Agricultural Engineering, Jilin University, 5988 Renmin Street, Changchun 130025, China; myh@jlu.edu.cn (Y.M.); wanghx17@mails.jlu.edu.cn (H.W.); qihy725@163.com (H.Q.); yujt18@mails.jlu.edu.cn (J.Y.)
2   The Key Laboratory of Bionic Engineering, Ministry of Education, Jilin University, 5988 Renmin Street, Changchun 130025, China
3   State Key Laboratory of Automotive Simulation and Control, Jilin University, Changchun 130022, China
*   Correspondence: zhuangjian_2001@163.com; Tel.: +86-1(874)-302-7198

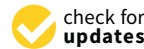

**Featured Application: This work proves that the bionic design method using badger teeth outlines can be applied to reduce the force and improve the soil penetration process, which can be further utilized for the optimization of tillage.**

**Abstract:** Soil penetration is the most important process during soil tilling. To optimize the soil penetration process, six specimens were designed and fabricated based on the badger teeth outlines. Both experimental investigation and numerical analysis were conducted with three types of soil. Results showed the specimen C, B, and D got the lowest penetration force and reduced the force by 26.15%, 22.68%, and 25.86% compared with that of specimen A under soil 1, soil 2, and soil 3, respectively. Depth-force curve analysis showed that the bionic specimens can slow down the force increase rate by reducing the coefficient of the force-depth curve equations. The bionic specimens obtained a lower increase of internal friction angle and cohesion after penetration, indicating the soil strength after penetration was lower. Furthermore, the rise in soil surface was observed after the penetration, and the penetration with the bionic specimens got a higher rise. Simulation analysis showed that the mechanism for the force reduction was because the force direction was changed, which brought a better flowability and less strength for the soil. It concludes that the badger teeth outlines reduce the penetration force by changing the force directions and optimizing the soil properties. Based on research results, the optimal bionic curve for penetration in different types of soil was determined.

**Keywords:** bionics; biomechanical; finite element modeling; soil properties; penetration force

## 1. Introduction

During agricultural production, soil tillage is the most key step and affects various aspects from plowing to harvest. The mechanization of agricultural tillage has allowed farmers to shake off the heavy manual plowing and planting works, which largely improve the crop production efficiency. So far, soil mechanized tillage is faced with many problems, especially the excessively large penetration resistance and large energy consumption of mechanical parts. Among all soil tillage process, the penetration process is the first step, and the most force-consuming process. Due to the rapid increase of energy-consuming, the force reduction is the most important research which need to be further researched [1].

Many studies have been conducted regarding the analysis of force during soil penetration. Onwualu [2] reviewed the soil-machine interaction in soil bin test facilities, which are useful for the design and performance evaluation for tillage tools. Bianchinia and Magalha [3] evaluated the soil performance of gear coulters through a series of experiments. Sahu and Raheman [4] predicted the design requirements of common tillage tools under any field conditions, according to the design requirements of tillage tools and soil properties under given soil conditions. Kushwaha et al. [5] studied that both the soil and tool parameters influence the penetrating forces. Al-kheer et al. [6] believed that the change of soil failure modes was the major factor of soil tool design. Based on previous researches, it was proved that the force was closely related to the soil properties and tool geometry. However, the optimization method to reduce the force has not been treated in much details, which is still a research hotspot.

Recently, bionic design has been regarded as a novel design method for the optimization of industries, such as the surface modification [7], bionic robots [8], and engine industries [9]. Meanwhile, the bionic design is also utilized for agricultural industries, for example, P. Ball [10] revealed the drag reduction mechanism in fluid medium. The excavating functions of mole claws offered good clues for designing anti-resistance force soil cutting instruments and excavation instruments [11]. Bo Li [12] analyzed the excellent property of bear claws for the future design of subsoiling tools, and found the best performance of the claw was at the rake angle of 30° with consideration into draft force and soil disturbance.

Of all the biological characteristics, the animal teeth have the best comparability with the penetrating process, regardless of the profiles and occluding modes [13]. The animal teeth normally have four forms: Incisor (cutting and biting), canine (piercing and tearing), premolar (chopping), and molar (squashing and chewing) [13]. Badgers have sharp teeth that can easily bite wires. Our previous study showed that the badger teeth have outstanding cutting and biting capabilities due to the special surface curves and high sharpness, and thus the teeth could penetrate materials with a lower force [14]. This extraordinary phenomenon is attributed to the occlusion modes and teeth outlines of badgers [15]. On basis of research on the biological characteristics of badger teeth, our project is aimed to explore bionic design theories and techniques for tillage. Thus, the excellent curves of badger teeth can be potentially selected as the bionic prototype to optimize the soil penetration process.

In this work, to optimize the penetration process and reduce the resistance, six different specimens were designed and fabricated based on the badger canine using bionic design method. Experimental investigation and numerical analysis of the penetration process for the cones with different curves were conducted to characterize and compare the penetration forces. At the same time the change of soil properties after penetration using different cones was compared. Moreover, the internal friction angle and cohesion of the soils were tested before and after experiments. Based on the research, the best bionic outlines for different types of soils was obtained and the mechanisms of the force reduction by the geometry of the cone was analyzed.

## 2. Materials and Methods

### 2.1. 3D Model Acquisition

Domestically raised badgers (Carnivora, Mustelidae, male) who normally lived in Northeast China were used in this study. The genetic properties of badgers established the whole incisor outlines, and the wear and friction process during eating forms the final outlines with excellent cutting properties [16]. The badger is a group living omnivore, and there are no hierarchies in feeding experiments within the groups [17]. So, there would be no difference in teeth outlines between the badgers in one group. A badger which is at the age of five years old and weighs 10 kg with a body length of 70 cm was selected as the prototype. The five years' feeding process provided enough time to render the teeth beneficial outlines by wear and friction [18]. During the sampling, the right upper canines were pulled out under complete anesthesia, which was conducted via standard intravenous

injection of 40 mg/kg pentobarbital sodium. Then the animals were diminished with inflammation and injected with an awakening agent. All animal welfare and experimental procedures were performed strictly in accordance with the Guidelines set by the Care and Use of Laboratory Animals (National Research Council of USA, 1996) and the related ethical regulations of Jilin University, China. All animal experiments were approved and guided by the Experimental Animal Management Center of Jilin University, China.

The obtained tooth as shown in Figure 1a was disinfected in anhydrous ethyl alcohol and then washed with distilled water for 3 min. After the surface residual alcohol was removed, the tooth was cleaned under vibration using a KQ-100E ultrasonic cleaner (Kun-Shan Ultrasonic Instruments Co., Ltd., Kunshan, Jiangsu Province, China) to remove the residual surface impurity. Then, the cleaned tooth was scanned using a Vivid910 3D laser scanner (Konica Minolta Co., Tokyo, Japan). The developing agent was sprayed on the surface before scanning to acquire a clear 3D point cloud. Sectional analysis and optimization modeling were implemented by GEOMAGIC reverse-engineering software. The exposed part of the tooth was selected, and the other part was ignored. Then the 3D point cloud in Figure 1b was imported into SOLIDWORKS 2016 to form a 3D model as shown in Figure 1c.

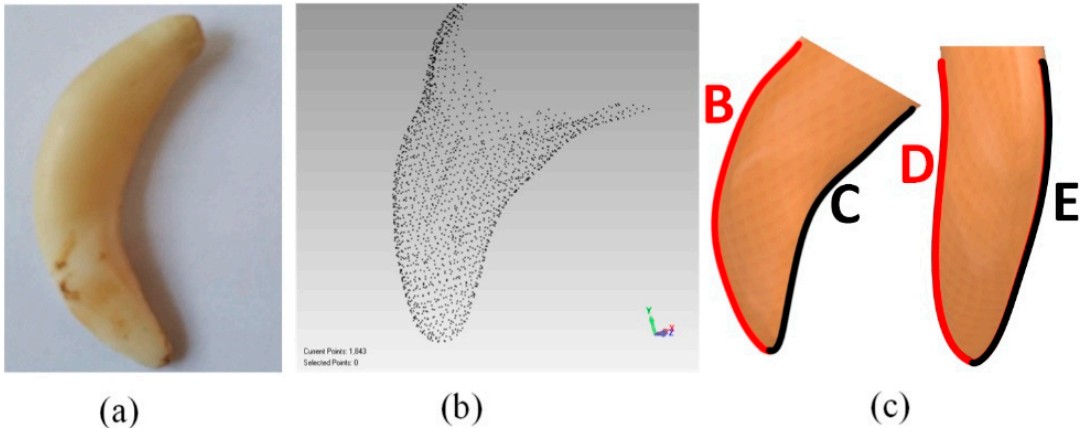

**Figure 1.** The badger canine. (**a**) The real badger canine. (**b**) The point cloud of the canine involving the front 15 mm. (**c**) The 3D model and selected outlines of badger canine.

## 2.2. Specimens Design and Fabrication

The function of canine during eating is penetration, and the outlines play an important role in the penetration process [19]. One tooth can be divided into four parts. The anterior and posterior surfaces of canines acted to penetrate into food, while the left and right surfaces functioned to fix and assist with the penetration [20]. To specifically design the applicable specimens for different type of materials, four different outlines marked as y2, y3, y4, and y5 from left to right as shown in Figure 1c was selected as the prototype. At the same time, a comparison curve y1 was created using a similar tendency of the bionic curves. The obtained outlines were magnified to a total length of 40 mm. The parametric equations of all the lines as shown in Figure 2a were fitted by MATLAB which can be represented in the below,

$$y1 = 2x^{0.5} \tag{1}$$

$$y2 = 0.04537 \times x^2 - 0.2926x^1 + 2.507x^{0.5} \tag{2}$$

$$y3 = 0.001839 \times x^2 - 0.05856x^1 + 1.709x^{0.5} \tag{3}$$

$$y4 = 0.001403 \times x^2 + 0.01259x^1 + 1.431x^{0.5} \tag{4}$$

$$y5 = 0.003 \times x^2 - 0.08676x^1 + 1.631x^{0.5} \tag{5}$$

where x and y are the horizontal and ordinate axes of the curve, respectively. The R-Square of each fitting was higher than 0.985 which can make sure the accuracy of the fitting.

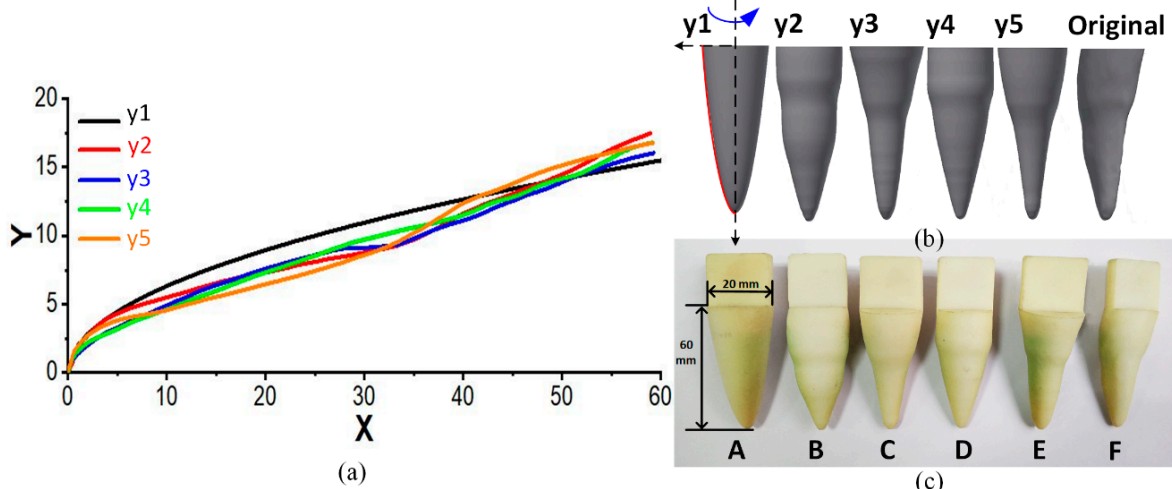

**Figure 2.** (**a**) The exposed canine outlines, (**b**) the designed 3D model built in SOILDWORKS, and (**c**) the fabricated specimen using 3D printing. In (**b,c**), A is the comparison specimen designed using y1, B–E are the bionic specimen designed using y2, y3, y4, and y5, and F is the original tooth surface specimen.

The given structure curves, with the apex as the center, was completely revolved once around the y-axis as shown in Figure 2b to form an entity structure, which was used as the specimen A to E. At the same time, the 3D specimen model of the original tooth surface was used as specimen F. All the models were created in SOLIDWORKS 2016 and shown in Figure 2b. The specimens had a ~30° cone which is in accordance with the American Society of Agricultural and Biological Engineers (ASABE) [21]. Based on the designing, the six specimens from A to F were imported via the fast molding processing technique into a ProJet5500X 3D printer (3D Systems Corporation, Rock Hill, SC, USA) for fabricating. The printing material was acrylonitrile-butadiene-styrene copolymer (ABS). It has an elastic modulus of $2 \times 103$ MPa, shear modulus of 318.9 MPa, mass density of $1.02 \times 103$ kg/m$^3$, tensile strength of 30 MPa, thermal conductivity of 0.2256 W/(m·K), and specific heat of 1386 J/(kg·K). The fabricated specimens were shown in Figure 2c.

### 2.3. Design and Preparation of Penetration Tests

Since the tools performance depend on different soil environments [22–25], three types of soils as shown in Figure 3a were selected, which were marked as soil 1, soil 2, and soil 3, respectively. The composition of the soils which were collected from Jilin University, China, was listed in Table 1. The soils were firstly screened over a 10-mesh vibrating sieve, dried, and adjusted by adding purified water. Shear strength parameters (cohesion and internal friction angle) of the soil were measured before and after the penetration tests using a direct shear apparatus [26]. The soils were put into 400 by 400 by 150 mm$^3$ box and compacted under 50 N pressure to be 110 mm thick for tests. At the same time, the properties of soils were tested and listed in Table 2. Since the boxes were far larger than the diameters of the bionic structures, the boundary effect can be ignored.

**Table 1.** The composition of experimental soils.

|          | Soil 1 | Soil 2 | Soil 3 |
|----------|--------|--------|--------|
| Sand (%) | 72.53  | 25.85  | 25.85  |
| Clay (%) | 20.68  | 53.99  | 53.99  |
| Silt (%) | 6.79   | 20.16  | 20.16  |

**Table 2.** Physical properties of experimental soils.

|  | Soil 1 | Soil 2 | Soil 3 |
|---|---|---|---|
| Bulk density (g/cm$^3$) | 1.98 | 1.53 | 1.22 |
| Moisture (%) | 0 | 0 | 25 |
| Cohesion (Kpa) | 9.8 | 13.5 | 10.2 |
| Internal friction angle (°) | 30.8 | 28.2 | 25.5 |

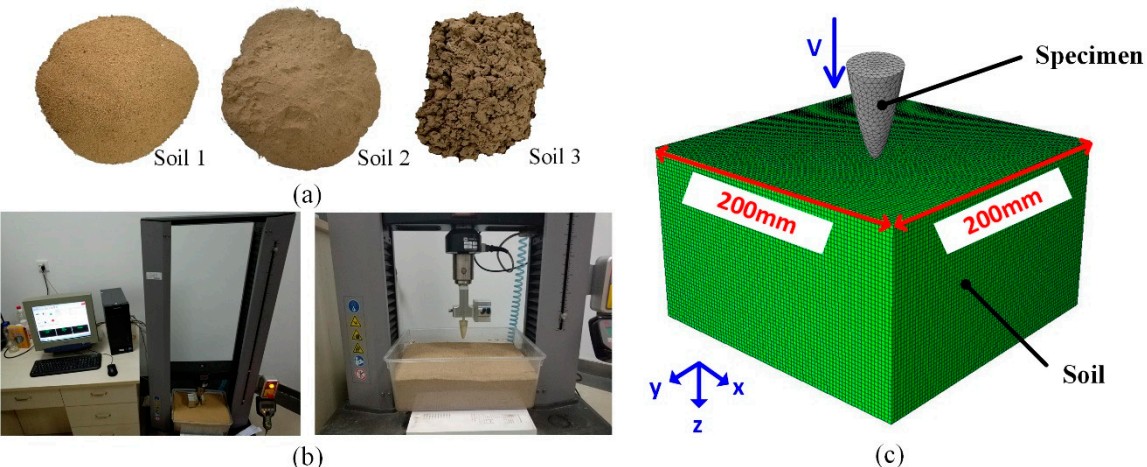

**Figure 3.** (**a**) The soils used in experiments. (**b**) The devices in experiments. (**c**) The simulation model for analyzing penetration process which was built in ABAQUS.

The force was measured using an all-purpose tester as shown in Figure 3b,c. The revolving models were clipped by a clamping device into the upper part of the all-purpose tester, and the processed soils were placed at the right lower part. The tests were conducted with the speed of 4 mm/s and falling height of 50 mm. The force of each specimen penetrating each type of soil was measured. Each test was repeated five times, and the average was calculated.

### 2.4. Finite Element Modeling

Agricultural soils experience plastic deformations after penetration induced by an engaging tillage tool. The behavior of the soil before yielding is elastic, but the elastic range is usually small and does not represent the real stress range [27] (i.e., the stresses applied by tillage tools are beyond the elastic range). Therefore, defining the soil material in an FEM model consists of importing both the elastic and plastic properties and selecting the yield and plastic strain behavior using constitutive models. The Drucker–Prager (D–P) model [28] was used in this study, it is a modified version of von Mises model, and considers the influence of hydrostatic pressure in failure. The extended D–P models are used to model frictional materials, such as soils and rock, where material yield is associated with hardening (i.e., the material strength increases with increasing stress level).

The D–P model is defined as follows:

$$F = t - p \times \tan \beta - d = 0 \tag{6}$$

where F is the yield function, t is the deviatoric stress, p is the normal stress, β is the internal friction angle, and d is the cohesion of the material. Moreover, p and t are given as follows [29]:

$$p = \frac{1}{3}(\sigma_1 + \sigma_2 + \sigma_3) \tag{7}$$

$$t = \frac{1}{2}q \left[1 + \frac{1}{k} - \left(1 - \frac{1}{k}\right) \left(\frac{r}{q}\right)^3\right] \tag{8}$$

$$q = (\sigma_1 - \sigma_3) \tag{9}$$

where k (flow stress ratio) is the ratio of tension yield stress to compression yield stress in triaxial tests (0.778 < k < 1). k and β were calculated as follows:

$$K = \frac{3 - \sin \varphi}{3 + \sin \varphi} \tag{10}$$

$$\tan \beta = \frac{6 \sin \psi}{3 - \sin \psi} \tag{11}$$

where φ is the internal friction angle and ψ is the Dilatancy angle.

The elastic modulus for each soil was determined by unconfined uniaxial compression test [30]. The Poisson's ration ν was calculated as follows:

$$v = \frac{1 - \sin \psi}{1 + (1 - \sin \psi)} \tag{12}$$

The soil cohesion and the angle of internal friction were measured through direct shear test. The soil property parameters used in FEM model were shown in Table 3.

**Table 3.** Soil properties of soils used in FEM model.

| Properties | Soil 1 | Soil 2 | Soil 3 |
|---|---|---|---|
| Density (mg/m$^3$) | 1.98 | 1.53 | 1.22 |
| E, Elastic modulus (Mpa) | 4.3 | 2.25 | 1.25 |
| ν, Poisson's ratio | 0.47 | 0.43 | 0.43 |
| K, Flow stress ratio | 0.93 | 0.89 | 0.85 |
| β, Internal friction angle (°) | 30.8 | 28.2 | 25.5 |
| ψ, Dilatancy angle (°) | 0.1 | 0 | 0 |
| C, Cohesion (Kpa) | 9.8 | 13.5 | 10.2 |
| f, coefficient of friction | 0.6 | 0.45 | 0.3 |

The 3D models were developed in ABAQUS/Explicit as shown in Figure 3. The soil model part was 200 mm long, 200 mm wide, and 100 mm deep. The axis of each specimen was vertical in the center of the soil. Each specimen was considered as a rigid body and meshed with R3D4 element type. And the soil was meshed with C3D8R element type, which is commonly used for 3D stress–strain analysis of continuum materials [31]. As reported, the mesh size importantly affects the tillage force and calculation time [1]. The soil model had a fine mesh with 307,200 elements which can ensure the accuracy of the results.

Meanwhile, Eulerian Lagrangian re-meshing coupled with mesh control was utilized to preserve a high-quality mesh of the soil model during the specimen penetration and prevent excessive mesh distortion [32]. This is an important feature in problems with high degree of material deformation. The interaction between specimen and soil was simulated with a general contact law and tangential behavior. A penalty contract method was used which searched for node-to-surface penetrations in the current configuration. The rigid body was selected as the master and the deformable part as the slave automatically by the software. Boundary conditions applied were: (1) both side walls of the soil (in y–z plane) were constrained; (2) the bottom face of the soil (in x–z plane) was constrained in both y and z directions to prevent movement of the soil whilst the top face of the box was left free of any constraint; (3) the end face of the box (in x–y plane) was fixed whilst the face in contact with the specimen was left with no constraint; and (4) the specimen was applied at a velocity of 4 mm/s on the reference point with a distance of 50 mm in the vertical direction (z direction). The outputs of the model were the force on the reference nodes (the plane on the root of the specimen) of the specimen. Further outputs of

interest were the stresses in the soil, some energy components of the model including kinetic energy, total internal energy, and elastic–plastic strain energies.

## 3. Results

### 3.1. Penetration Force

Figure 4a showed the simulation model and the maximum force of each specimen penetrating different soils. Specimen A got the maximum force for all the types of soil. For the other specimens, the resistance was lower than that of specimen A in varying degrees, indicating the bionic curves have ability in reducing resistance during penetration, and the bionic specimens designed by different teeth curves positions have different performances in one soil. Moreover, the specimens C, B and D obtained the lowest force under soils 1, 2, and 3, respectively. Take specimen A into the basis, the maximum force of optimal specimen was reduced by 26.15%, 22.68%, and 25.86%, respectively. Thus, it is believed the bionic specimens C, B, and D had the lowest penetration force in penetrating soils 1, 2, and 3, respectively, which should be selected to conduct the further analysis.

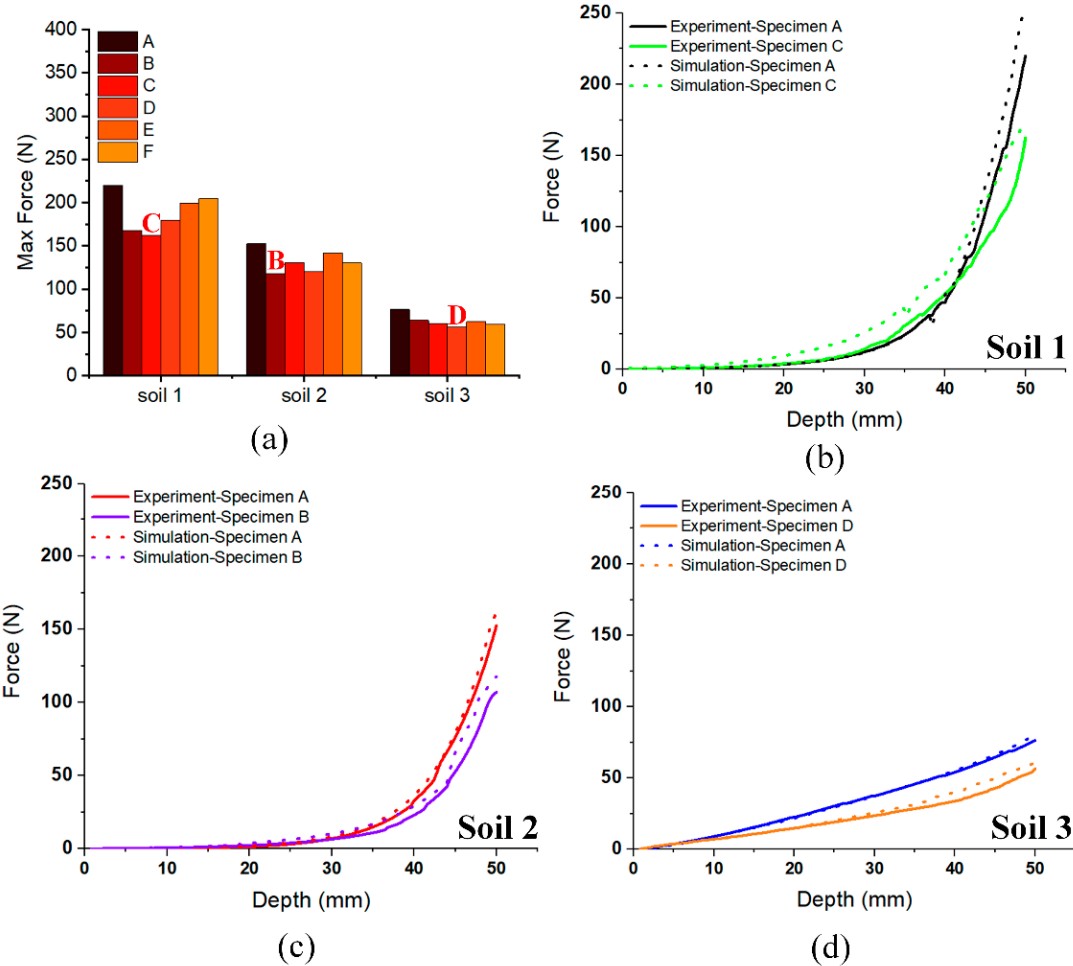

**Figure 4.** (**a**) The max force of experiments and; The depth-force curves of (**b**) specimen A and C penetrating soil 1, (**c**) specimen A and B penetrating soil 2, and (**d**) specimen A and D penetrating soil 3 in experimental and simulation results.

From Figure 4b–d, the depth-force curves during the penetration using specimen A and C in soil 1, specimen A and B in soil 2, and specimen A and D in soil 3 in the simulation is compared with that in the experiments. It proves that the depth-force curve compared well, which can verify that the

experimental tests and numerical simulations were in close agreement. The increase tendency of force curves was similar in soil 1 and soil 2, while it had distinct difference when comparing with soil 3.

In order to analysis the mechanism of the reduction in different soils, the force equations of the specimens with maximum and minimum final force in three soils were fitted by MATLAB. During the penetration process, the curve of force in soil 1 and soil 2 can be described by $y = ab^x$, and the correlation was listed in the Equations (13)–(16). The R-Square of the equations are higher than 0.95, which accord with the accuracy.

$$Y_{A1} = 0.1458 \times 1.158^x \ (R-Square \ = \ 0.9987) \tag{13}$$

$$Y_{C1} = 0.5716 \times 1.119^x \ (R-Square \ = \ 0.9991) \tag{14}$$

$$Y_{A2} = 0.07495 \times 1.165^x \ (R-Square \ = \ 0.9902) \tag{15}$$

$$Y_{B2} = 0.4986 \times 1.116^x \ (R-Square \ = \ 0.9634) \tag{16}$$

Indeed, b is the main parameter that can represent the increase rate of the force. As b in Equations (6) and (8) was smaller compared with Equations (13) and (15), respectively, which means it can slow down the force increase rate, in that way the reduction of penetration force can be enhanced in the end. Moreover, the larger a in Equations (14) and (16) counteracted the influence of b, which lead to the phenomenon that the force difference at the beginning was small.

When looking at how the specimens penetrated the soil 3, the force-depth curves can be described by $y = cx - d$ as shown in Equations (17) and (18). The R-Square of the equations are also higher than 0.95, which accord with the accuracy. It is illustrated the force was increased linearly, while for soil 1 and soil 2 the increase method was index.

$$Y_{A3} = 1.528x - 6.154 \tag{17}$$

$$Y_{D3} = 0.9915x - 3.531 \tag{18}$$

The specimen D got the smaller c, which is the slope of the function, representing the increase rate of force. Same as the soil 1 and soil 2, the mechanism of the force reduction when using the bionic outline was because the bionic outline can reduce the c and slow down the force increase rate.

### 3.2. Soil Properties before and after Penetration

According to Coulomb's theory [33], internal friction angle and cohesion are the main influence factors on soil strength, which related to the penetration force [34]. Therefore, the internal friction angle and cohesion of the three soils were tested before and after experiments. In Figure 5a,b the internal friction angle and cohesion before penetration in soils 1 and 2 are larger than soil 3, as higher water content reduces the bonding and frictional resistance between the soil particles [35,36]. Moreover, the internal friction angel and cohesion after penetration were increased modestly with the three soils, which was because the soil strength should be increased after the penetration [37]. Meanwhile, after penetration using different specimens, the internal frictional angle and cohesion was distinctly different. It is illustrated that the internal friction angle and cohesion after penetration using bionic specimen was smaller than that of using specimen A. According to Coulomb's laws, lower cohesion and internal friction angle lead to the higher soil strength. The higher soil strength means the soil was more difficult to be penetrated, which resulted in a higher penetration force. Thus, the mechanism for the lower force using bionic specimen was because that the bionic curves can reduce the increase of internal friction angle and cohesion during the penetration process, while the soil strength after penetration was also lower.

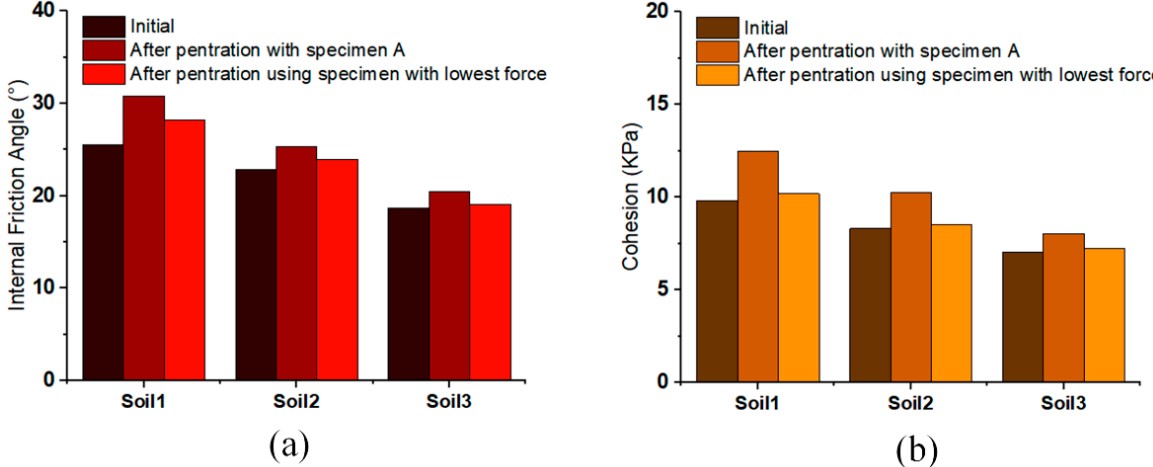

**Figure 5.** (**a**) The internal friction angle cohesion and (**b**) the cohesion of the soils before and after tests.

### 3.3. Soil Surface Status after Penetration

Figure 6a,b showed the soil status at the end of tests for the specimens with maximum and minimum force in both experimental and simulation results. During the penetration, rise occurred at the surface of soil 1 and soil 2 around the specimens. However, there was no obvious rise in soil 3, and the sinking was found in the soundings of the specimen. When talking about the rise height, the specimen C under soil 1 and specimen B under soil 2 was higher than that of specimen A under the same soil. Evident sinking (a radian on the soil surface close to the specimen) with soil 3 was found in specimen A, but not in specimen D, indicating the soil rise got by specimen D neutralized the sinking. It means that the better flowability of the soil was obtained when penetrating using bionic specimen. Conversely, it also means that the soil suffered more pressure and got higher soil strength when penetrating using specimen A. It can thus be assumed that the reason for the lower force obtained by bionic specimen was because the soil had more flowability and less soil strength during the test, which made the soil more easily penetrated.

Figure 6c shows the force analysis during specimen penetration. The different surface rise level was obtained using different specimens, indicating the directions of force was different as well as soil moving status. According to Newton's Second Law, the crosswise force $F_x$ and the portrait force $F_y$ imposed on the soils were:

$$F_x = F21 + F22 = F \times \cos \varphi \tag{19}$$

$$F_y = F11 - F12 = F \times \sin \varphi \tag{20}$$

A larger proportion of $F_y$ results in the higher rise for soil, and a larger proportion of $F_x$ subjected the soils to more pressure and more soil strength, which was unfavorable for the penetration. According to Equations (19) and (20), the $F_x$ and $F_y$ are related to the included angle $\varphi$. between the forces and the horizontal direction; larger $\varphi$ leads to smaller proportion of $F_x$ and larger proportion of $F_y$; and vice versa.

To analysis the force in soil, the force direction imagines was obtained by simulation as shown in Figure 6d. The obtained $\varphi$ was listed in Table 4. It believes that, no matter what kind of soils, the bionic specimen had a larger $\varphi$ and then got a larger proportion for $F_y$ and a smaller proportion for $F_x$, indicating the bionic specimen did change the directions of forces imposed on the soils during penetration. The larger proportion of $F_y$ and smaller proportion of $F_x$ when using the bionic specimen leaded to the better flowability and lower soil strength, which is the mechanism for the higher rise for soils and lower penetration force.

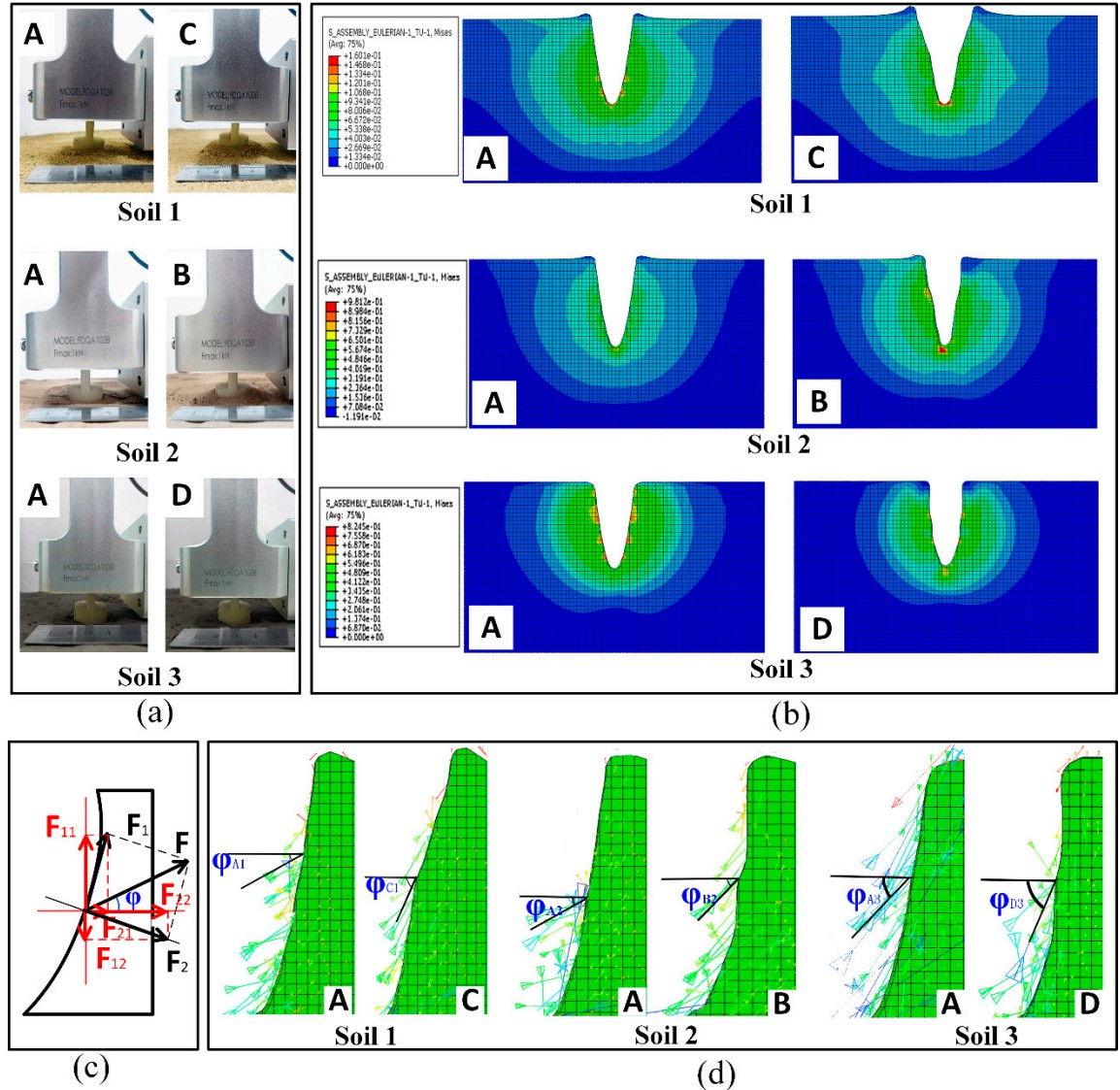

**Figure 6.** (**a**) Soil status after the experiments for the specimens with maximum and minimum force. (**b**) Stress results in soil simulations for the specimens with maximum and minimum force in soil 1, soil 2, and soil 3. (**c**) Force analysis during specimen penetration in soil 1, soil 2, and soil 3. (**d**) Vector-graphs of forces in soil 1, soil 2, and soil 3.

**Table 4.** The direction of penetration force.

| φ | Specimen A | Specimen C in Soil 1, Specimen B in Soil 2 and Specimen D in Soil 3 |
|---|---|---|
| Soil 1 | 29.5 | 61.8 |
| Soil 2 | 30.1 | 45.3 |
| Soil 3 | 47.2 | 67.1 |

## 4. Conclusions

In this work, the penetration process was studied in three different types of soils with six specimens designed and fabricated based on the badger canine outlines. Several main conclusions can be drown from this research as follows:

- Geometry of cone affects the penetration force substantially. Experimental results showed that the bionic specimens C, B, and D obtained the lowest penetration force in soils 1, 2, and 3, respectively.
- The lower penetration force resulted from bionic specimens (specimens C, B, and D) should be attributed to the fact that their soil strength after penetration was lower than that by using specimen A.
- The simulation results showed that the force direction was changed using bionic specimen, resulting in a higher proportion of vertical force, a lower proportion of horizontal force, and a better flowability, which is also part of the reason for the lower penetration force.

The bionic design using badger teeth outlines can be applied to reduce the penetration force and improve the soil penetration process in tillage. This work is an attempt to apply badger teeth outlines for the bionic design of tillage tools. A limitation of this study is that the real tillage tool was not fabricated, and the field test was not conducted. Further work will focus on establishing the bionic design for the real tillage tools.

**Author Contributions:** Data curation, H.W., H.Q. and J.Y.; Formal analysis, H.W.; Funding acquisition, Y.M. and J.Z.; Methodology, H.W.; Project administration, Y.M.; Resources, Y.M.; Software, H.W. and J.Y.; Supervision, Y.M. and J.Z.; Validation, J.Z. and H.Q.; Writing—Original draft, H.W., H.Q. and J.Y.; Writing—Review & editing, Y.M., H.W. and J.Z. All authors have read and agree to the published version of the manuscript.

**Funding:** This study was funded by the State Key Laboratory of Automotive Safety and Energy (No. KF1814), National Natural Science Foundation of China (No. 51475205), Department of Science and Technology of Jilin Province (Nos. 20170101173JC and 20170204015NY), National Key Research Program of China (Nos. 2016YFD0701601 and 2017YFD0701103-1), Jilin Provincial Development and Reform Commission (No. 2018C044-3), China-EU H2020 FabSurfWAR project (Nos. 2016YFE0112100 and 644971), and the 111 Project of China (No. B16020), Jilin Province education office "the thirteenth five-year plan" industrialization project (JJKH20180076KJ), and China Scholarship Council (CSC201806170191).

**Conflicts of Interest:** The authors declare no conflict of interest.

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
