# Peer review of "Effects of Bionic Curves on Penetration Force under Difference Soils"

_applsci, doi:10.3390/app10020529_

Round 1

Reviewer 1 Report

Dear Authors, 

I have read your manuscript, and I think it may be recommended for publication without much modification. At the same time, the text should be carefully checked to eliminate minor typos. Also in the introduction it is necessary to formulate your hypothesis more clearly.

Sincerely yours, reviewer

Author Response

Response to Reviewer 1 Comments

Point: I have read your manuscript, and I think it may be recommended for publication without much modification. At the same time, the text should be carefully checked to eliminate minor typos. Also, in the introduction, it is necessary to formulate your hypothesis more clearly.

Response: The authors appreciate the valuable comments from the reviewer. The manuscript has been carefully checked, and all typos have been removed. The introduction has also been modified to better formulate the hypothesis.

Reviewer 2 Report

The paper: “Effects of bionic curves on penetration force during vertical penetration under difference soils”

The title of the paper might be adjusted. (Two same words penetration,    …maybe  “  different soils”)

 “Curves” or “geometry of cone” have impact on penetration force?

 Line 20. “Results showed the specimen C, B and D got the lowest force and reduced the force by 26.15%, 22.68% and 25.86% under soil 1, soil 2 and soil 3, respectively.”

It should be explained result of research; which force has decreased. It should be explained that differences of force is relative to compare specimen A.

The aim of this study was…    continue …

Research developed …  new cone geometry… continue ..

Based on research results, the optimal bionic curve…was determined …  continue…

 Introduction part is reasonable.  

Line 71 “Experimental investigation and numerical analysis of cone penetration process were conducted to characterize and compare the forces and the soil properties.”

Need to explain in more detail what forces are being investigated and with whom have they been compared?

Line 75 “mechanisms of the force reduction was analyzed.“ - Need to explain in more detail what mechanisms? Or only geometry of cone?

 In “Materials and Methods” part maybe not to explain so much about that animal torture.

Line 85   „the right upper canines were pulled out “– horrible, an unfortunate animal.

Can’t take the photo of teeth? After reading such an article, the scientists will be desired to pull out the  tails of elephant's.

Which the selected outline of badger canine from Figure 1 „(c) the 3D model and selected outlines of badger canine “ for B C D of fabricated specimens? Need to explain in more detail.

Line 140 – “ 400x300x150 mm3  - Mistake

Table 1 Wet soil? maybe – clay. Soil classifications are important in determining several building and landscaping limitations. The system of classifying soil is especially important when it comes to laying foundations. All soils react to changes in moisture, therefore the soil on which you build your foundations must offer enough support.

Line 154 “Agricultural soils experience plastic deformations after yielding induced by an engaging tillage tool.”  - what is the yield?

Line 164 “where F is the yield function“  - the “yield”, it is appropriate word?

“tension yield stress to compression yield stress”

Line 168 “where φ is the M-C internal friction angle and ψ is the internal angle of friction - is it the same?

ψ, Dilatancy angle (°)

Line 192 The outputs of the model were the reaction forces on the reference nodes of the specimen. - with what “The force was measured using an all-purpose tester” the force was measured, maybe you can explain.

Line 204.  it thus believed the bionic specimen C, B and D had the best performance in penetrating soil 1, 2 and 3, - grammar mistake.

The pressure force (Figure 4), is the main indicator of paper. Why is the pressure force increasing so dramatically at depth from 40mm to 50mm, (soil 1 and soil 2) if the soil is homogeneous?

Line 231 „The specimen D got the smaller c, which resulted for the lower force. Same as the soil 1 and soil 2, the mechanism of the force reduction when using the bionic outline was because the bionic outline can reduce the c and slow down the force increase rate.”

Need to explain what “c”. Need more detail about force increase “rate”

Line 288. Table 4. The value of ? in Figure 9.  – Mistake       

„Comparison specimen“  -  new term in paper – maybe „speciment A“

“Bionic specimen”

 Conclusions must be improved, the findings more clearly described.

The conclusions - the main short points of paper, not to repeat exactly what you have written before.

Concluding statements – from title of the paper:

What effects of bionic curves?

What the penetration force? increase/decrease, (in numerical meaning) compares it with comparison specimen A (in numerical meaning), during soil penetration on different soils – (according soil classification Clay, Loam, Silt, Sand, et. all)

The conclusions are an answer to question of problem. The conclusions are mine part of research. The conclusions must rewrite. The text of conclusions so long, is difficult to understand the notion.

Line 293 “The multiple regression analysis indicates the bionic specimen can reduce the increase speed of the force during penetration.” – what “speed”? It is new term.

Line 296 “Then, rise on the soil surface was more significant using bionic specimens after the penetration. –where measurements of rise in paper?

Line 297 “That is because the bionic curve indeed changed the direction of the force and got a higher proportion of Fy  and a lower proportion of Fx, which lead to a better flowability and less strength for soil.” – discussion of results part.

Line 299 “All the discussion prove that the bionic design method using badger teeth outlines can be applied to reduce the force and improve the soil penetration process in tillage, and the different sides of the outlines can be applied for different types of soil.“

 Repeated sentences of discussions. Repeat sentences without any specific penetration force meaning. Where solutions of bionic design of cone geometry to the penetration force?

Author Response

Response to Reviewer 2 Comments

The authors appreciate the reviewer’s valuable comments.

The edited part in the manuscript has been highlighted with yellow color.

Detailed response for each question can be seen on the below:

Point 1: The paper: “Effects of bionic curves on penetration force during vertical penetration under difference soils” The title of the paper might be adjusted.  (Two same words penetration …maybe “different soils”) “Curves” or “geometry of cone” have an impact on penetration force?

Response 1: The title has been changed to ‘Effects of bionic curves on penetration force under difference soils.’.

The geometry of cone, which was designed by the bionic curves, had the main impact on penetration force.

Point 2: Line 20. “Results showed the specimen C, B and D got the lowest force and reduced the force by 26.15%, 22.68% and 25.86% under soil 1, soil 2 and soil 3, respectively.”

It should be explained result of research; which force has decreased. It should be explained that differences of force are relative to compare specimen A.

The aim of this study was…    continue …

Research developed …  new cone geometry… continue ...

Based on research results, the optimal bionic curve…was determined …  continue…

Response 2: The details of force reduction have been explained as recommended. The aim of the study has been better presented at the end of the abstract.

Point 3: Line 71 “Experimental investigation and numerical analysis of cone penetration process were conducted to characterize and compare the forces and the soil properties.” Need to explain in more detail what forces are being investigated and with whom have they been compared?

Response 3: The penetration force was being investigated and the change of soil properties after penetration using different cones was compared. The detailed explanation of the comparation was added.

Point 4: Line 75 “mechanisms of the force reduction was analyzed.“ - Need to explain in more detail what mechanisms? Or only geometry of cone?

Response 4: The mechanism of force reduction by different geometry of cone was analyzed. The detail was added in the manuscript.

Point 5:  In “Materials and Methods” part maybe not to explain so much about that animal torture.

Line 85   „the right upper canines were pulled out “– horrible, an unfortunate animal. Can’t take the photo of teeth? After reading such an article, the scientists will be desired to pull out the tails of elephant's.

Which the selected outline of badger canine from Figure 1 „(c) the 3D model and selected outlines of badger canine “ for B C D of fabricated specimens? Need to explain in more detail.

Response 5: The declaration of animal torture proved that the animal was treated with painless, which was in accordance with the Guidelines set by the Care and Use of Laboratory Animals.

The canine was firstly analyzed for the outlines in this research, and then the other research such as the materials composition and the microstructure will be analyzed. So, it must be pulled out. At the same time, the canine of the badger is only 1-2 cm outside the gum, which is very small. It may result in a big error if using the profile data obtained from the picture. The other factor is that the badger is fed artificially, which means losing one canine is not very harmful to its life.

The selected outlines for the different cones were marked in Figure 1c as recommended. 

Point 6: Line 140 – “400x300x150 mm - Mistake

Table 1 Wet soil? maybe – clay. Soil classifications are important in determining several building and landscaping limitations. The system of classifying soil is especially important when it comes to laying foundations. All soils react to changes in moisture, therefore the soil on which you build your foundations must offer enough support.

Response 6: The mistake was corrected to 400x400x150 mm3. The word ‘wet soil’ in Table 1 was changed to ‘clay’. 

Point 7: Line 154 “Agricultural soils experience plastic deformations after yielding induced by an engaging tillage tool.”  - what is the yield?

Response 7: ‘Yield’ does not most accurately express the meaning in this sentence. Therefore, it is changed to ‘penetration’.

Point 8: Line 164 “where F is the yield function” - the “yield”, it is appropriate word?

“tension yield stress to compression yield stress”

Response 8: The phrase ‘yield function’ in this sentence should be correct, since it can be found in the following references:

He C, You Y, Wang D, Wang G, Lu D, Kaji JM. The effect of tine geometry during vertical movement on soil penetration resistance using finite element analysis. Computers and electronics in agriculture. 2016 Nov 15;130:97-108. Naderi-Boldaji M, Alimardani R, Hemmat A, Sharifi A, Keyhani A, Tekeste MZ, Keller T. 3D finite element simulation of a single-tip horizontal penetrometer–soil interaction. Part I: Development of the model and evaluation of the model parameters. Soil and Tillage Research. 2013 Nov 1;134:153-62. Armin A, Fotouhi R, Szyszkowski W. On the FE modeling of soil–blade interaction in tillage operations. Finite elements in analysis and design. 2014 Dec 1;92:1-1.

Point 9: Line 168 “where φ is the internal friction angle and ψ is the internal angle of friction - is it the same? ψ, Dilatancy angle (°)

Response 9: The ψ is the Dilatancy angle, and the mistake in the manuscript has been corrected.

Point 10: Line 192 The outputs of the model were the reaction forces on the reference nodes of the specimen. - with what “The force was measured using an all-purpose tester” the force was measured, maybe you can explain.

Response 10: The output should be the force (rather than reaction force) on the reference nudes of the specimen, which is accorded with the simulation setup. The word ‘reaction’ in the sentence was deleted.

Point 11: Line 204.  it thus believed the bionic specimen C, B and D had the best performance in penetrating soil 1, 2 and 3, - grammar mistake.

The pressure force (Figure 4), is the main indicator of paper. Why is the pressure force increasing so dramatically at depth from 40mm to 50mm, (soil 1 and soil 2) if the soil is homogeneous?

 Response 11: The grammar mistake was corrected.

The force increased dramatically from 40s to 50s because the soil suffered more pressure due to the larger volume in the roof of the cone.  In that way the soil strength was increased sharply.

Point 12: Line 231 „The specimen D got the smaller c, which resulted for the lower force. Same as the soil 1 and soil 2, the mechanism of the force reduction when using the bionic outline was because the bionic outline can reduce the c and slow down the force increase rate.”

 Need to explain what “c”. Need more detail about force increase “rate”

Response 12: ‘c’ is the slope of the depth-force function. The force increase rate can be represented by this variable. More details were added in the manuscript.

Point 13: Line 288Table 4. The value of ? in Figure 9.  – Mistake       

„Comparison specimen“  -  new term in paper – maybe „specimen A“

“Bionic specimen”

Response 13: The title and the top row of Table 4 was changed.

Point 14: Conclusions must be improved, the findings more clearly described.

The conclusions - the main short points of paper, not to repeat exactly what you have written before.

Concluding statements – from title of the paper:

What effects of bionic curves?

What the penetration force? increase/decrease, (in numerical meaning) compares it with comparison specimen A (in numerical meaning), during soil penetration on different soils – (according soil classification Clay, Loam, Silt, Sand, et. all)

The conclusions are an answer to question of problem. The conclusions are mine part of research. The conclusions must rewrite. The text of conclusions so long, is difficult to understand the notion.

Response 14: The conclusion part was rewritten to better summarize this research.

Point 15: Line 293 “The multiple regression analysis indicates the bionic specimen can reduce the increase speed of the force during penetration.” – what “speed”? It is new term.

Response 15: In the previous conclusion, the word ‘speed’ should be changed to ‘rate’, However,

a new conclusion section has been formulated, and this sentence has been rewritten.

Point 16: Line 296 “Then, rise on the soil surface was more significant using bionic specimens after the penetration. –where measurements of rise in paper?

Response 16: The rise on the soil surface was observed in the picture after experimental penetration in Figure 6a.  And the exact difference was compared using simulation results as shown in Figure 6b. It is not necessary to measure the value of rise because this study only focuses on the difference of results between each specimen during penetration.

Point 17: Line 297 “That is because the bionic curve indeed changed the direction of the force and got a higher proportion of Fy  and a lower proportion of Fx, which lead to a better flowability and less strength for soil.” – discussion of results part.

Response 17: This part indeed is a summary of the results, it should not show up in the conclusion section. Therefore, it is removed from the conclusion section.

Point 18: Line 299 “All the discussion proves that the bionic design method using badger teeth outlines can be applied to reduce the force and improve the soil penetration process in tillage, and the different sides of the outlines can be applied for different types of soil. “

 Repeated sentences of discussions. Repeat sentences without any specific penetration force meaning. Where solutions of bionic design of cone geometry to the penetration force?

Response 18: This sentence should be removed from the conclusion. The results showed that the bionic specimen C, B, and D obtained the lowest penetration force in Soil 1, 2 and 3, respectively, and thus leading to the best performance.
